# 210-W, Quasi-Continuous Wave, Nd:YAG InnoSlab Laser at 1319 nm

**Xiangrui Meng [1,2,3], Xingwang Luo [1,2], Junzhi Ye [1,2], Xiaoming Chen [1,2,\*], Xuguang Zhang [1,2], Lei Zhang [1,2], Qingsong Gao [1,2] and Baole Lu [3]**

1    Institute of Applied Electronics, China Academy of Engineering Physics, Mianyang 621900, China; mengxiangruijm@163.com (X.M.); zhuchuelxw@126.com (X.L.); 15349841308@163.com (J.Y.); zhang_131517@163.com (X.Z.); lakylei@163.com (L.Z.); 15883798199@163.com (Q.G.)
2    Key Laboratory of Science and Technology on High Energy Laser, Chinese Academy of Engineering Physics, Mianyang 621900, China
3    State Key Laboratory of Photon-Technology in Western China Energy, Institute of Photonics and Photon Technology, Northwest University, Xi'an 710127, China; lubaole1123@163.com
\*    Correspondence: chenxiaomingzs@163.com

**Abstract:** In this paper, we demonstrate a high-power, quasi-continuous wave using a laser-diode dual-end-pumped Nd:YAG InnoSlab laser at 1319 nm. The maximum average output power of 210 W at a single 1319 nm wavelength is obtained with an optical-optical efficiency of 18.8% from absorbed pump power to laser output. The output pulse duration is 246 μs at the repetition of 500 Hz, and the beam quality factors of $M^2$ are 1.37 and 1.47 in the horizontal and vertical directions, respectively. This is the first report on high-power, quasi-continuous wave using Nd:YAG InnoSlab lasers at 1319 nm with good beam quality.

**Keywords:** quasi-continuous wave; InnoSlab; Nd: YAG; 1319 nm





## 1. Introduction

Since Nd:YAG material exhibits excellent optical characteristics, it has been commonly used in many high-power laser systems [1–5]. Beside the well-known laser operation at 1064 nm by Nd:YAG, high-power 1319 nm lasers have been extensively studied for their excellent applications. For instance, they can be used in surgery [6], optical fiber communication [7], color display [8,9], and converted into red and yellow light by non-linear frequency conversion [10–12]. However, the effective stimulated cross section emission for transition of 1319 nm is $8.7 \times 10^{-20}$ cm², which is only one fourth of that of the 1064 nm wavelength. In addition, the 1319 nm laser operation has a much larger quantum defect. Therefore, it is much more difficult to obtain a high-power laser output at 1319 nm. In 2007, a diode-side-pumped Nd:YAG rod continuous wave (CW) laser at 1319 nm was described [13]. The output power was 131 W under a pump power of 555 W, with an optical-optical conversion efficiency of 23.6% and a beam quality factor of $M^2$ of ~51. In 2022, T.Y. Zang et al. reported a 808 nm laser-diode (LD) end-pumped Nd:YAG slab laser at 1319 nm by using a plano-concave stable cavity [14]. The CW output power of 109 W was obtained with an optical-optical efficiency of 20.9% and $M^2$ beam quality factors of 850 and 3.06 in the horizontal and vertical directions, respectively. Beside the above CW laser operation, the latter at 1319 nm can also be operated in quasi-continuous wave (QCW) mode. In 2017, Chuan Guo et al. obtained a 51.5 W QCW 1319 nm output by using a Nd:YAG slab laser amplifier with the extraction efficiency of 14.2%. The beam quality factors were $M^2_x = 1.61$ and $M^2_y = 1.81$ [15]. Although these QCW laser operations have remarkable good beam quality, the average output power needs a further scale. Until now, achieving high-power laser operation at 1319 nm with good beam quality has proved challenging with traditional laser geometries.

Partially end-pumped slab (InnoSlab) lasers are characterized by the stable-unstable hybrid resonator, which can effectively alleviate thermal effects. Compared to traditional solid-state lasers, InnoSlab lasers have higher power output and better beam quality due to their cavity design and optical coupling characteristics [16,17]. Recently, researchers have shown that both InnoSlab amplifier and resonator schemes are able to generate 1319 nm laser radiations [18–20]. In 2013, a high-power, high-beam-quality, diode end-pumped Nd:YAG InnoSlab amplifier at 1319 nm was reported. In a five-pass configuration, the amplifier yielded a 42.3 W output with an optical-optical efficiency of 6.5% and beam quality factors of $M^2_x$ = 1.13 and $M^2_y$ = 2.16 in the orthogonal directions [19]. In 2022, H.L. Zhang et al. described a Nd:YAG InnoSlab laser resonator at 1319 nm with a maximum output power of 23.2 W, an optical-optical efficiency of 11.36% and a slope efficiency of 17.8%. How to scale the output power and optical conversion efficiency while maintaining good beam quality still requires further research [20].

In this paper, we demonstrate an LD dual-end-pumped, Nd:YAG InnoSlab laser for high-power QCW operation at 1319 nm. A maximum average output power of 210 W at a single 1319 nm wavelength is reported for the first time with an optical-optical efficiency of 18.8%, a slope efficiency of 22.9%, and $M^2$ beam quality factors of 1.37 and 1.47 in the horizontal and vertical directions, respectively.

## 2. Experimental Setup and Principle

Figure 1 shows the structure of the QCW 1319 nm InnoSlab laser resonator. The gain medium used in the experiment is a Nd:YAG slab crystal with dimensions of 24 mm (a) ×1 mm and (b) ×12 mm (c). The Nd$^{3+}$-ions concentration is 1 at.%. Two QCW 808 nm LD arrays are employed as the pump source (the red line indicates the 808 nm pump light, and the yellow line indicates the 1319 nm light). Each LD array is composed of 2 × 10 LD bars which are vertically encapsulated and collimated with micro-lenses in the fast axis direction. The coupling system includes some cylindrical lens. The pump light is focused tightly into the planar waveguide (8 mm × 30 mm × 80 mm) to make the beam more uniform in the slow axis direction. Two polarizers (P1, P2) and a half waveplate (HWP) in the coupling systems are used to protect the LDs from the residual radiation of the LDs on the other side. The LD pump light is injected into the crystal through two end surfaces with a uniform distribution of 24 mm width in the slow-axis direction and a Gaussian distribution of 0.4 mm width in the fast-axis direction. The two large surfaces of the crystal are welded by Indium with two heat sinks of copper, dissipating waste heat through the water-cooled copper and reducing the crystal temperature, which can help improve the laser output power and beam quality.

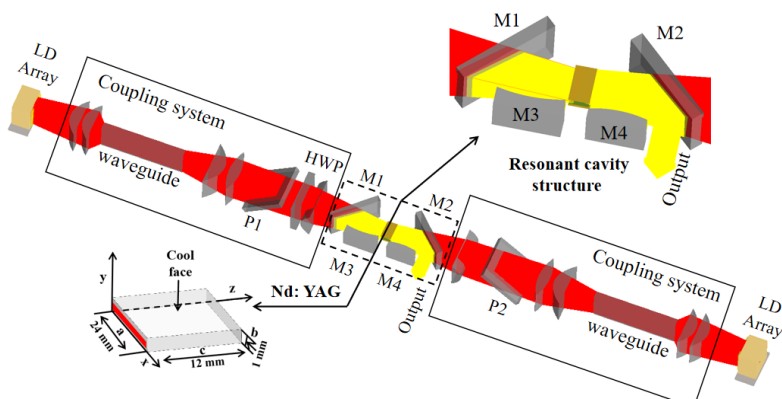

**Figure 1.** Schematic of the LD dual-end-pumped Nd:YAG InnoSlab laser resonator.

M1 and M2 are 45° mirrors with an anti-reflection (AR) coating at 808 nm and 1064 nm and high-reflection (HR) coating at 1319 nm, and they serve the purpose of folding the resonant cavity. M3 and M4 are concave and convex cylindrical mirrors with a radius of

curvature of $R_1$ = 700 mm and $R_2$ = −500 mm, respectively. They are used as the resonant cavity mirrors, and are both coated with 1319 nm HR and 1064 nm AR. The geometric distance between the two cavity mirrors is 100 mm, forming a positive-branch confocal unstable resonator in the 24 mm width direction of the Nd:YAG crystal. The laser beam in an "unconfining" or unstable resonator diverges away from the axis and eventually radiation will spill around the edges of the M4 edge. The equivalent transmittance of the resonant cavity is determined by the curvature of the two cavity mirrors, with the expression of equivalent coupling transmittance being T = 1 − | $R_2$ |/$R_1$ = 28%.

Figure 2a shows the intensity distribution presented in the Nd:YAG crystal after the 808 nm LD pump light passes through the coupling systems. It can be seen that the pump light is generally a rectangular spot with a uniform distribution in the horizontal direction. In the vertical direction, the intensity distribution shows a Gaussian pattern. In addition, the total coupling efficiency of the pump light is about 90% and the absorption efficiency by the crystal is ~80% in the experiments. Figure 2b shows the temperature rise estimation of the crystal under a maximum absorbed pump power of 1197 W, and the maximum temperature can be controlled at about 77.1 °C. No large temperature fluctuation is found under long-time operation, indicating that the cooling system is normal and supports the stable operation of the 1319 nm laser. The concentration level of the Nd:YAG crystal can be reduced to alleviate a future temperature rise in the crystal [21,22].

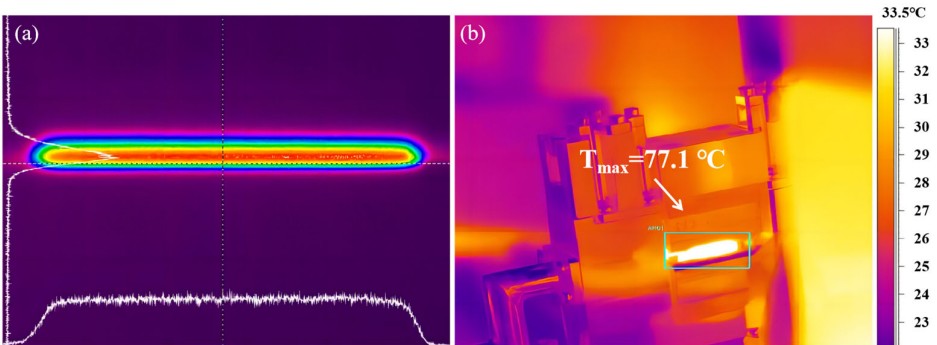

**Figure 2.** (**a**) Pump beam intensity distribution in the Nd:YAG crystal and (**b**) temperature rise estimation of the crystal at the absorbed pump power of 1197 W.

## 3. Results and Discussion

In our experiments, the two QCW 808 nm LD arrays and the Nd:YAG crystal were all cooled by circulating water at a temperature of 25 °C. The repetition rate of the pump light was set to 500 Hz and the average output power is shown in Figure 3 for the pump pulse widths of 150 µs, 200 µs, 250 µs and 300 µs. It can be seen that a maximum average output power as high as 248 W is obtained at the pump pulse width of 300 µs, with a corresponding optical-optical efficiency of ~20.1% and an oscillator threshold power of 205 W. When the average output power is 210 W, the optical-optical efficiency and slope efficiency are 18.8% and 22.9%, respectively. To the best of our knowledge, this is the first report on Nd:YAG InnoSlab laser at 1319 nm with such a high output power and efficiency.

For the output power of 248 W, the power fluctuation is measured and shown in Figure 4. The average output power of the laser in 30 min is recorded by a power meter, and the power fluctuation is measured to be ±2%. As the measurement time increases, the laser output power decreases linearly. The reason for this phenomenon is that during the operation of the laser, the mirror holder of the resonant cavity continuously absorbs stray light, causing the holder to heat up and resulting in a slight decrease in laser output power. By optimizing the mirror holder structure in the future, the cooling and heat dissipation effect can be helpful to reduce the impact of temperature on it.

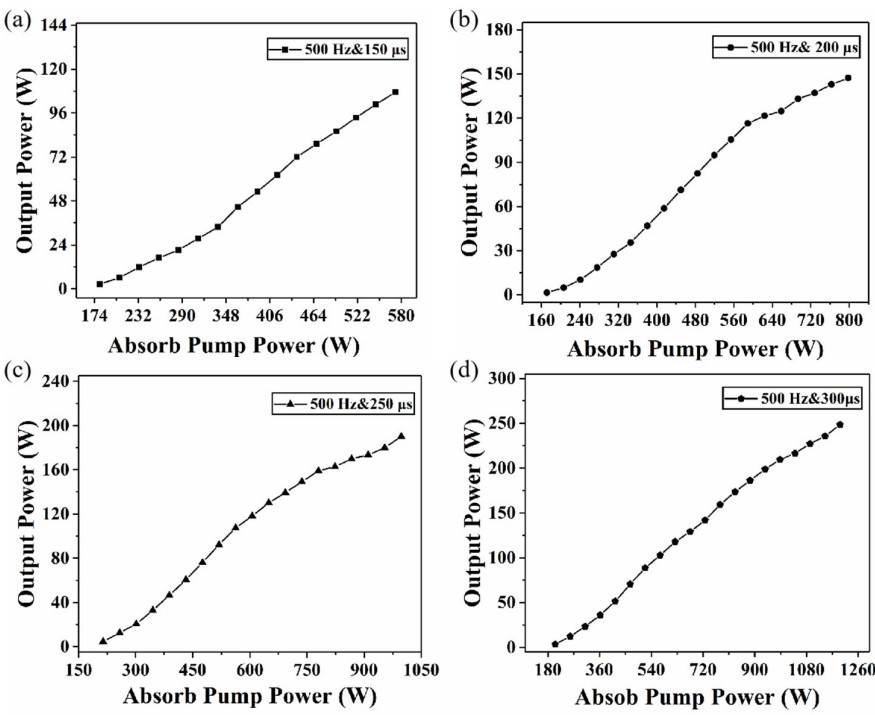

**Figure 3.** Laser output power versus absorbed pump power. (**a**) 500 Hz and 150 μs; (**b**) 500 Hz and 200 μs; (**c**) 500 Hz and 250 μs; (**d**) 500 Hz and 300 μs.

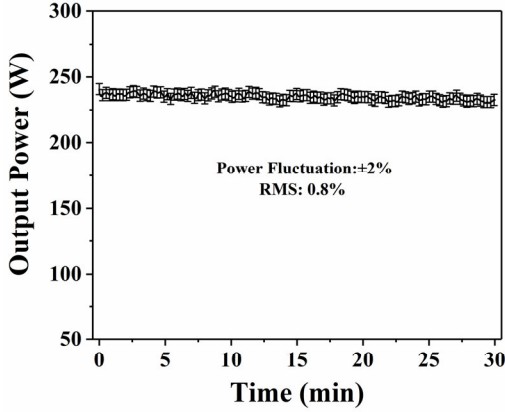

**Figure 4.** Power stability measurement.

In addition, the laser output wavelength is detected by a spectrometer (Yokogawa (Tokyo, Japan), AQ6370D). Figure 5a,b depict the measured spectra at the average output power of 210 W and 248 W, respectively. The spectral property of the output beam is scanned and analyzed from 1000 nm to 1400 nm. As it can be seen from Figure 5a, we obtain 1319 nm QCW laser operation with a single wavelength when the output power is below 210 W and the subfigure in Figure 5a represents the narrow-range spectral shape captured at the single wavelength, whilst increasing the pump power further causes the 1338 nm wavelength to oscillate, as shown in Figure 5b. This is because the laser gain medium has two strong transitions in the 1.3 μm region from the $^4F_{3/2}$ to the $^4I_{13/2}$ manifold which have almost the same stimulated emission cross section. Thus, the laser output typically contains both wavelengths. One is the R2 → X1 transition at 1319 nm and the other is the R2 → X3 line at 1338 nm. We can further reduce the reflectivity of the cavity mirrors (M3 and M4) at 1338 nm to suppress its oscillation [15].

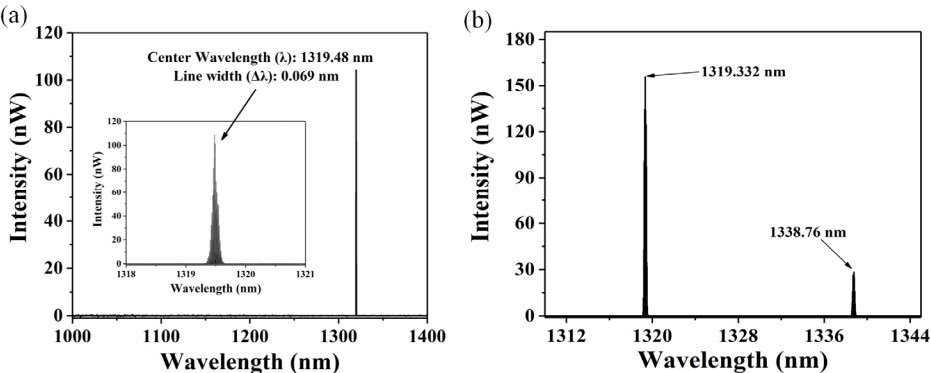

**Figure 5.** Measured spectra at the output power of 210 W and 248 W, respectively.

When the laser operates at the average output power of 210 W, the output pulse duration and the pulse profile are detected by a photodetector and a digital oscilloscope (Tektronix (Beaverton, OR, USA), DPO7354). As depicted in Figure 6a,b, the output pulse repetition rate and the pulse width are 500 Hz and 246 μs, respectively. The pulse waveform is smooth, which indicates that the power extraction is sufficient and there is no self-excitation in the 1319 nm InnoSlab laser.

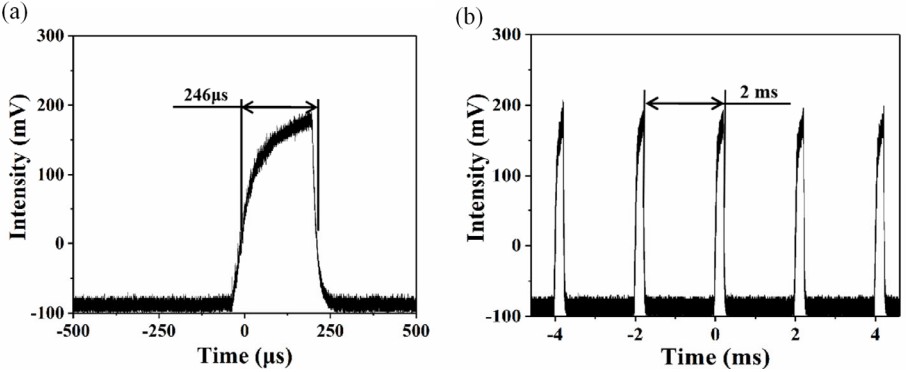

**Figure 6.** (**a**) Single pulse waveform and (**b**) pulse train.

Next, for the average output power of 210 W, the two-dimensional profile of the laser beam is measured by a beam quality analyzer (Spiricon, $M^2$-200), as shown in Figure 7. The beam quality factors of $M^2$ with the method of 4-sigma are 1.37 and 1.47 in the horizontal and vertical directions, respectively.

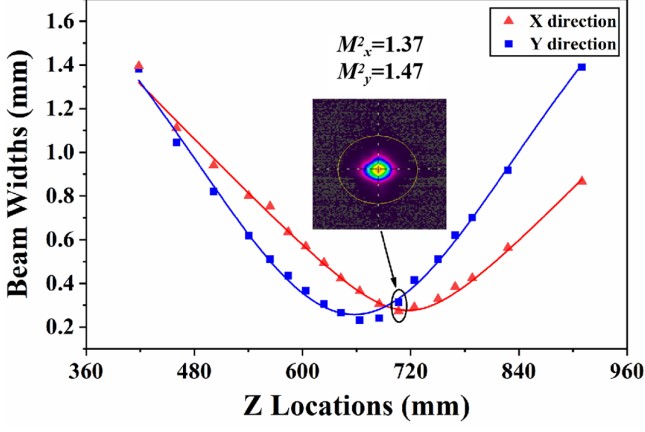

**Figure 7.** Beam quality measured after a beam shaping system.

## 4. Conclusions

In summary, we demonstrate a high-power, quasi-continuous wave using an Nd:YAG InnoSlab laser emitting a single wavelength at 1319 nm. When the 808 nm pump pulse duration is 300 μs at the repetition rate of 500 Hz, a maximum average output power of 210 W is achieved and the output laser pulse width is 246 μs. The optical conversion efficiency is 18.8% and the slope efficiency is 22.9%. The corresponding beam quality factors of the horizontal axis and the vertical axis are $M^2_x = 1.37$ and $M^2_y = 1.47$, respectively.

**Author Contributions:** Conceptualization, X.C. and X.L.; methodology, X.C. and X.L.; software, X.C. and X.M.; validation, X.M. and X.C.; formal analysis, X.M. and X.C.; investigation, X.Z., X.M., B.L., L.Z. and J.Y.; resources, X.C. and Q.G.; data curation, X.C. and X.L.; writing—original draft preparation, X.M. and X.C.; writing—review and editing, X.M. and X.C.; visualization, X.M. and X.C.; supervision, X.M. and X.C.; project administration, X.C. and X.L. All authors have read and agreed to the published version of the manuscript.

**Funding:** Presidential Foundation of CAEP (Grant No. YZJJLX2019015).

**Institutional Review Board Statement:** Not applicable.

**Informed Consent Statement:** Not applicable.

**Data Availability Statement:** The raw data supporting the conclusions of this article will be made available by the authors, without undue reservation.

**Conflicts of Interest:** The authors declare no conflict of interest.

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
