# Peer review of "210-W, Quasi-Continuous Wave, Nd:YAG InnoSlab Laser at 1319 nm"

_photonics, doi:10.3390/photonics10070730_

Round 1
Reviewer 1 Report
The subject of this manuscript is interesting and the obtained results deserve to be published. In connection with the content of the manuscript, some comments follow.
1. In the introduction, the authors mention a rather large number of references, namely [6-14], for which no discussion is presented. Alternatively, references can be given after each application, for example “surgery [x], military [x], and so on.
2. As references, most of the works presented are by authors from China! However, numerous other references can be found, some being pioneering works in the field (laser emission at 1.3 microns in Nd:YAG). In the end, it is the authors' decision whether or not to improve / modify the references section.
3. Figure 1 contains other inset figures, which are not numbered and discussed in the text. For example, the dimensions of the Nd:YAG laser medium (a, b, c) must be explained in the text; what each inset figure represents must be mentioned. The authors are asked to improve the section dedicated to the description of the experimental setup.
4. Lines 64 - 65, it says: “The two large surfaces of the crystal are welded by Indium, dissipating …”. Probably the authors want to say that those two surfaces are in contact with Indium; or that Nd:YAG was wrapped in Indium! Please use the most appropriate description!
5. Figure 3. Output power is presented as a function of diode current. However, in the text the discussion is based on the pump power! The reviewer’ recommendation is that Figure 3 shows the laser power versus on the pump power absorbed in Nd:YAG. Then, the discussion of this figure, which contains many data, is done on the run, without insisting on some characteristics! This section could be improved.
6. In experiments the authors use 1.0-at.% Nd:YAG. Why was this Nd doping level chosen? There are works [for example Opt. Express 13 (20), 7948-7953 (2005), or Appl. Phys. B 82 (04), 599-605 (2006)] that discusses the influence of Nd doping on thermal effects. It is clearly shown that for the emission at 1.3 microns in Nd:YAG with 1.0-at.%, the medium heats up compared to the case where there is no laser emission. This aspect can be a (partial) justification for the temperature indicated in Fig. 2(b) of Nd:YAG! The authors could add comments in the manuscript related to these aspects!
7. The method used to determine the M2 factor (Figure 6) should be indicated, or the device / apparatus used should be mentioned! Was the "knife-edge" method used?
Based on these comments, the reviewer's opinion is that the manuscript needs clarifications and improvements before being published. The reviewer's recommendation is "Major Revision".
Improvements in English are absolutely necessary.
Author Response
Response Letter
We sincerely thank the editor and all reviewers for their valuable comments. The reviewers’ comments are laid out below in italicized fonts. Our responses are given in normal fonts and changes/additions to the manuscript are given in red fonts in the revised manuscript.
Responses to Reviewer #1:
- In the introduction, the authors mention a rather large number of references, namely [6-14], for which no discussion is presented. Alternatively, references can be given after each application, for example “surgery [x], military [x], and so on.
Response:
The article has been amended to separate the references.
- As references, most of the works presented are by authors from China! However, numerous other references can be found, some being pioneering works in the field (laser emission at 1.3 microns in Nd:YAG). In the end, it is the authors' decision whether or not to improve / modify the references section.
Response:
Thanks to the reviewer's comments, some other references have been added.
- Figure 1 contains other inset figures, which are not numbered and discussed in the text. For example, the dimensions of the Nd:YAG laser medium (a, b, c) must be explained in the text; what each inset figure represents must be mentioned. The authors are asked to improve the section dedicated to the description of the experimental setup.
Response:
The description of other inset figures in Figure 1 has been supplemented.
- Lines 64 - 65, it says: “The two large surfaces of the crystal are welded by Indium, dissipating …”. Probably the authors want to say that those two surfaces are in contact with Indium; or that Nd:YAG was wrapped in Indium! Please use the most appropriate description.
Response:
The sentence from lines 64-65 in the article has been rephrased.
- Figure 3. Output power is presented as a function of diode current. However, in the text the discussion is based on the pump power! The reviewer’ recommendation is that Figure 3 shows the laser power versus on the pump power absorbed in Nd:YAG. Then, the discussion of this figure, which contains many data, is done on the run, without insisting on some characteristics! This section could be improved.
Response:
Figure 3 (a) has been changed to the curve of absorption pump power and output power, and has been modified in the paper.
- In experiments the authors use 1.0-at.% Nd:YAG. Why was this Nd doping level chosen? There are works [for example Opt. Express 13 (20), 7948-7953 (2005), or Appl. Phys. B 82 (04), 599-605 (2006)] that discusses the influence of Nd doping on thermal effects. It is clearly shown that for the emission at 1.3 microns in Nd:YAG with 1.0-at.%, the medium heats up compared to the case where there is no laser emission. This aspect can be a (partial) justification for the temperature indicated in Fig. 2(b) of Nd:YAG! The authors could add comments in the manuscript related to these aspects!
Response:
I have already included a discussion on the Nd3+ concentration in the article and have cited the two references suggested above. The Nd doping level can be reduced for alleviating the temperature rising in the crystal in the future.
- The method used to determine the M2 factor (Figure 6) should be indicated, or the device / apparatus used should be mentioned! Was the "knife-edge" method used?
Response:
Figure 6 shows the beam quality factors measured using a beam quality analyzer (Spiricon, M2-200). The measurement method employed is the 4-sigma method within the instrument system.
Reviewer 2 Report
The comments and questions to the paper are as follows.
1) The Figure 1 and its description can be improved: the full pumping beam transmitted through the planar waveguide, but the figure showed the partial transmission. What were the waveguide dimensions?
2) 1064 nm laser operation was discriminated by the mirrors M1-M4 antireflecting at 1064 nm. What residual reflection did these mirrors have at 1064 nm?
3) What spectrophotometer was used?
Minor editing of English language required
Author Response
Responses to Reviewer #2:
1) The Figure 1 and its description can be improved: the full pumping beam transmitted through the planar waveguide, but the figure showed the partial transmission. What were the waveguide dimensions?
Response:
The pump light is indeed fully incident into the waveguide. The figure showed the partial transmission is the pump beam distribution on the side of the crystal, not the waveguide.
2) 1064 nm laser operation was discriminated by the mirrors M1-M4 antireflecting at 1064 nm. What residual reflection did these mirrors have at 1064 nm?
Response:
The resonator mirrors M3 and M4 are coated with high-transmission films for 1064 nm, thereby preventing oscillation of 1064 nm within the cavity. By observing the spectrum of the output laser using a spectrometer, no 1064 nm laser emission was detected. Utilizing a beam splitter for 1064 nm/1319 nm, the laser power at 1064 nm was observed using a power meter, and no laser output was detected. This indicates that the laser system did not generate a 1064 nm laser.
3) What spectrophotometer was used?
Response:
The following are the model numbers of the instruments used in the article (added to the article) :
Oscilloscope (Tektronix, DPO7354)
Beam Quality Analyzer (Spiricon, M2-200)
Optical Spectrum Analyzer (Yokogawa, AQ6370D)

Reviewer 3 Report
The paper from Xiangrui Meng et al. demonstrates a realization of a quasi-CW Nd: YAG InnoSlab laser at 1319 nm with an output power of 210W with an M^2 of 1.37(x) and 1.47 (y). They show the construction and discuss the main properties of their laser. Finally, they also explain the observation of the additional 1338.76 nm laser line. The paper is written very well and does not let too much space for the referee to criticize. I found some points which can be improved, and I do not understand them.
When I understand it correctly, the laser is pumped by two LD arrays (808 nm), one from the right side and the other from the left side. The mirrors M1 et M2 are AR pour 808 nm and 1064 nm, but HR for 1319 nm. The cavity is given by the distance between M3 and M4. Both mirrors (M3 and M4) are curved. The laser output, however, passes on the right side of mirror M4. There I have a problem understanding the beam's pathway and the cavity's construction. Can you please show the inside of the cavity on a separate graphic?
Figure 2 is well done, but the temperature scale is too small. Maybe you can make it somehow bigger. You can make the numbers more prominent and show only every second number.
Please start a new line for : …. 4.Discussion
Figure 3B Please put error bars on the points. Please check the precision of your power meter.
Figure 4. Do you have measured the 248W for both lines together with a standard power meter? If yes, it might be a good idea to calculate with the spectrum in (b) the percentage of power belonging to 1319.332 nm (Maybe, if you like, you can make a reference to the spectrum analyzer you are using.)
Write the atomic states correctly 4_F_3/2 and 4_I_13/2. I hope they allow you to have subscripts and superscripts in the text. If not, then ignore my comment.
Author Response
Responses to Reviewer #3:
1) When I understand it correctly, the laser is pumped by two LD arrays (808nm), one from the right side and the other from the left side. The mirrors M1 and M2 are AR pour 808 nm and 1064 nm, but HR for 1319 nm. The cavity is given by the distance between M3 and M4. Both mirrors (M3 and M4) are curved. The laser output, however, passes on the right side of mirror M4. There I have a problem understanding the beam's pathway and the cavity's construction. Can you please show the inside of the cavity on a separate graphic?
Response:
The internal diagram shown below represents the resonator cavity, which consists of a hybrid configuration of a stable cavity and an unstable cavity. It is one of the characteristics of an InnoSlab Laser. Therefore, during the laser output, the light passes on the right side of mirror M4 (characteristic of the unstable cavity). This cavity design helps achieve better beam quality for the output laser. The red line is 808 nm light and the yellow line is 1319 nm light.

2)Figure 2 is well done, but the temperature scale is too small. Maybe you can make it somehow bigger. You can make the numbers more prominent and show only every second number.
Response:
I have already redrawn Figure 2, ensuring clear representation of the temperature scale.
3)Figure 3B Please put error bars on the points. Please check the precision of your power meter.
Response:
Figure 3(b) has been redrawn with the addition of error bars.
4)Figure 4. Do you have measured the 248W for both lines together with a standard power meter? If yes, it might be a good idea to calculate with the spectrum in (b) the percentage of power belonging to 1319.332 nm (Maybe, if you like, you can make a reference to the spectrum analyzer you are using.)
Response:
Due to the absence of a 1319 nm/1338 nm beam splitter, the power of the two wavelengths was not individually measured using a power meter. However, it can be observed from the spectrometer that the power ratio between the two wavelengths is approximately 6:1.
5)Write the atomic states correctly 4_F_3/2 and 4_I_13/2. I hope they allow you to have subscripts and superscripts in the text. If not, then ignore my comment.
Response:
The superscripts and subscript in the article has been modified.

Round 2
Reviewer 1 Report
The authors provided satisfactory answers to the comments I made in the report. For the final version of this article, the following observations can be considered.
1. Line 29: “… is 8.7×10^(-20) cm^(2), which …”.
2. Line 73: “The pump light is focused thickly into …”! “thickly” or “tightly”?
3. Figure 3(a). Fewer experimental points are now presented! Certainly more results have a stronger impact! However, in the final it is the decision of the authors!
Finally, I recommend this manuscript for publication.
A final check of English is still recommended!
Author Response
- Line 29: “… is 8.7×10^(-20) cm^(2), which …”.
Response:
It has been modified to the superscript.
- Line 73: “The pump light is focused thickly into …”! “thickly” or “tightly”?
Response:
It has been modified as “tightly”.
- Figure 3(a). Fewer experimental points are now presented! Certainly more results have a stronger impact! However, in the final it is the decision of the authors!
Response:
More results have been added in Figure 3.

Reviewer 3 Report
Various correction has been done and increased the quality of the paper. I still don’t understand the cavity shown in fig1 of the publication. Normally a resonator is a closed arrangement of mirrors with one mirror acting as a coupling mirror. I have problems understanding where the amplified beam is leaving the resonator without having trouble having a closed loop. Figure 1 has to be improved or further explications have been given in the text.
Author Response
1) Various correction has been done and increased the quality of the paper. I still don’t understand the cavity shown in fig1 of the publication. Normally a resonator is a closed arrangement of mirrors with one mirror acting as a coupling mirror. I have problems understanding where the amplified beam is leaving the resonator without having trouble having a closed loop. Figure 1 has to be improved or further explications have been given in the text.
Response:
We have provided additional explanations for Figure 1 in the text. For the laser output mentioned in the paper, it can refer to the book 'Solid State Laser Engineering,' the section on unstable resonators.
Figure 1 is the structure diagram of the unstable resonator. Light ray in an "unconfining" or unstable resonator diverges away from the axis, and eventually radiation will spill around the edges of one or both mirrors.

Fig. 1 Light ray in an unstable resonator
Figure 2 illustrates the spherical waves bouncing between the mirrors of an unstable resonator. In this so-called confocal configuration, the concave mirror's focal point coincides with the back focal point of the smaller convex mirror. Thus we see that P1 is the common focus. Light traveling to the left is collimated and may conveniently be coupled out by providing an unobstructed aperture around mirror M1.

Fig.2 Confocal unstable resonator
The unstable resonator corresponds to a divergent periodic focusing system, whereby the beam expands on repeated bounces to fill the entire cross-section of at least one of the laser mirrors. The unstable resonator has a much higher order mode discrimination, as compared to its stable counterpart, and therefore a nearly diffraction-limited output beam from a large diameter gain medium can be obtained. In the near field, the output from an unstable resonator usually has an annular intensity pattern.

Round 3
Reviewer 3 Report
All suggestions have been realized in this paper. This why I think I this paper can be published.